# Impacts of High Environmental Temperatures on Congenital Anomalies: A Systematic Review

**DOI:** 10.3390/ijerph18094910

**Published:** 2021-05-05

**Authors:** Marjan Mosalman Haghighi, Caradee Yael Wright, Julian Ayer, Michael F. Urban, Minh Duc Pham, Melanie Boeckmann, Ashtyn Areal, Bianca Wernecke, Callum P. Swift, Matthew Robinson, Robyn S. Hetem, Matthew F. Chersich

**Affiliations:** 1The Heart Centre for Children, The Children’s Hospital at Westmead, The University of Sydney, Sydney 2041, Australia; marjan.mosalmanhaghighi@health.nsw.gov.au; 2Environment and Health Research Unit, South African Medical Research Council, Pretoria 0001, South Africa; Caradee.Wright@mrc.ac.za; 3Department of Geoinformatics, Geography and Meteorology, University of Pretoria, Pretoria 0001, South Africa; 4The Heart Centre for Children, The Children’s Hospital at Westmead, Westmead 2041, Australia; julian.ayer@health.nsw.gov.au; 5Clinical School, The Children’s Hospital at Westmead, The University of Sydney, Sydney 2041, Australia; 6Division of Molecular Biology and Human Genetics, Faculty of Medicine and Health Sciences, University of Stellenbosch, Parow, Cape Town 7505, South Africa; urban@sun.ac.za; 7Burnet Institute, Melbourne 3004, Australia; minh.pham@burnet.edu.au; 8Department of Epidemiology and Preventive Medicine, Faculty of Medicine Nursing and Health Science, Monash University, Melbourne 3004, Australia; 9Department of Environment and Health, School of Public Health, Bielefeld University, 33699 Bielefeld, Germany; boeckmannmelanie@gmail.com; 10IUF-Leibniz Research Institute for Environmental Medicine in Düsseldorf, 40225 Düsseldorf, Germany; ashtynareal@yahoo.com; 11Environment and Health Research Unit, South African Medical Research Council, Johannesburg 2094, South Africa; Bianca.Wernecke@mrc.ac.za; 12Environmental Health Department, Faculty of Health Sciences, University of Johannesburg, Johannesburg 2094, South Africa; 13Emergency Department, Tallaght University Hospital, D24 NR04 Dublin, Ireland; swiftcp@tcd.ie; 14Department of Public Health, Health Service Executive, R95 A002 Kilkenny, Ireland; drmatthewrobinson77@gmail.com; 15School of Animal, Plant and Environmental Sciences, Faculty of Science, University of the Witwatersrand, Johannesburg 2000, South Africa; Robyn.Hetem@wits.ac.za; 16Wits Reproductive Health and HIV Institute, Faculty of Health Sciences, University of the Witwatersrand, Johannesburg 2000, South Africa

**Keywords:** birth defects, congenital, climate change, environmental health, heat, neonates, maternal health

## Abstract

Links between heat exposure and congenital anomalies have not been explored in detail despite animal data and other strands of evidence that indicate such links are likely. We reviewed articles on heat and congenital anomalies from PubMed and Web of Science, screening 14,880 titles and abstracts in duplicate for articles on environmental heat exposure during pregnancy and congenital anomalies. Thirteen studies were included. Most studies were in North America (8) or the Middle East (3). Methodological diversity was considerable, including in temperature measurement, gestational windows of exposure, and range of defects studied. Associations were detected between heat exposure and congenital cardiac anomalies in three of six studies, with point estimates highest for atrial septal defects. Two studies with null findings used self-reported temperature exposures. Hypospadias, congenital cataracts, renal agenesis/hypoplasia, spina bifida, and craniofacial defects were also linked with heat exposure. Effects generally increased with duration and intensity of heat exposure. However, some neural tube defects, gastroschisis, anopthalmia/microphthalmia and congenital hypothyroidism were less frequent at higher temperatures. While findings are heterogenous, the evidence raises important concerns about heat exposure and birth defects. Some heterogeneity may be explained by biases in reproductive epidemiology. Pooled analyses of heat impacts using registers of congenital anomalies are a high priority.

## 1. Introduction

Globally, an estimated 3% of newborns have a congenital anomaly, which accounts for an estimated 295,000 newborn deaths each year according to the World Health Organization (WHO). Several fold more children have long-term disabilities, with major impacts on individuals, families, and health care systems [1,2]. The WHO figures likely underestimate the total burden as they exclude spontaneous or induced abortions and stillbirths related to congenital anomalies. Moreover, undercounting is likely in resource-constrained settings where surveillance systems for congenital anomalies are poorly developed. 

Most congenital anomalies arise from the process of malformation during intrauterine life, defined as “a non-progressive, congenital morphologic anomaly of a single organ or body part due to an alteration of the primary developmental program” [3]. Isolated organ malformations are typically considered multifactorial in aetiology, where an individual is affected if the combination of predisposing factors in both the genetic and the prenatal environment exceed a certain threshold [4]. Malformations may, however, also result from a single insult, either a specific genetic disorder or a teratogen, such as alcohol. 

While several environmental factors have been clearly implicated in teratogenesis, the potential harms of many environmental exposures remain uncertain. In particular, the potential teratogenicity of exposure to high ambient temperatures warrants consideration, with several factors indicating that such links are plausible. Firstly, maternal heat exposure has been strongly associated with a range of other pregnancy complications, such as preterm birth and stillbirths, suggesting an overall sensitivity of pregnancy to heat exposure [5]. Secondly, a large number of animal studies have documented causal links between high temperatures and birth defects, in studies as far back as 1877 [6,7,8,9]. Thirdly, maternal fever, internal heat generated from moderate-to-vigorous prenatal exercise, hot tubs, saunas and electrical blankets have been linked with birth defects in a number of studies, though these associations are contested [10,11,12,13]. Lastly, exposure to some teratogens may increase during extreme heat. Most notably, alcohol consumption generally rises during hot weather, and is especially hazardous during embryogenesis as this usually occurs prior to pregnancy diagnosis and the onset of alcohol aversion from morning sickness.

Extreme heat exposure is one of the largest threats to health in the 21st century [14], with global temperatures already 1.2 °C higher than in the pre-industrial period [15], and heatwaves are increasing in frequency, intensity, and duration. High temperatures pose particular dangers for groups with reduced thermoregulatory ability, such as pregnant women [7,16]. Moreover, in many countries, pregnant women lack resources to adapt to extreme heat and any emerging health risks in this population are a major concern given the already high risks for maternal and child deaths that these women face [17]. Given the above concerns, we systematically reviewed evidence of associations between heat exposure and the incidence of congenital anomalies. Unlike other systematic reviews which address teratogenic impacts of heat from maternal fever or hot spas, for example [6,18,19], this review focuses on ambient or environmental heat exposures. The findings are important for assessing the hazards that climate change poses for pregnant women, which remain underappreciated [20]. 

## 2. Methods

On 9 September 2018, we searched Medline (PubMed), and on 23 September 2018, we searched Science Citation Index Expanded, Social Sciences Citation Index, and Arts and Humanities Citation Index to identify articles on heat exposure and health outcomes. The literature search strategies are presented in Appendix A. As described in full elsewhere [5], in a first step, we screened titles and abstracts to identify studies that either assessed the effect of heat exposure on health outcomes or evaluated the effectiveness of adaptation interventions to reduce these impacts. Then, in a second step, the titles and abstracts of all studies fulfilling these criteria were screened to identify articles that reported on associations between heat in pregnancy and at least one congenital anomaly outcome. An additional search using simplified search terms was done on Medline (PubMed) on 13 December 2019 to locate articles published since the date of the initial search. We reviewed the full text of all articles reporting on heat exposure and pregnancy outcomes. Reference lists of included articles were also screened. 

Screening of titles, abstracts and full text articles was done independently in duplicate with any differences reconciled by a third reviewer or the principal investigator (MFC). We only included studies on humans, published in English, German, or Italian, and on heat related to weather rather than heat from sources such as saunas and hot baths. Studies that examined the impact of seasonality alone and not temperature *per se* were excluded. No date restrictions were applied. All study designs were eligible except for systematic reviews, and studies that modelled the potential or hypothetical associations between heat exposure and the review outcomes. EPPI-Reviewer version 4 software (Social Science Research Unit of the UCL Institute of Education, London, The United Kingdom) [21] provided a platform for management of articles; screening of titles, abstracts, full text articles; and for data extraction. 

Data were extracted in duplicate from eligible studies by M.M.H. and C.Y.W., including information on study characteristics (e.g., country and study periods), study methods such as design and temperature exposures, (measures of temperature and timing of exposure) and study outcomes (point estimates of associations and confidence intervals). We did not assess study quality, but we highlighted particular concerns around study quality as relevant. 

We report findings by type of defect, namely: congenital heart defects; neural tube defects; orofacial clefts or craniofacial defects; ocular anomalies; renal anomalies; hypospadias; musculoskeletal defects; congenital hypothyroidism; and any lethal anomaly. These anomalies are predominately organ malformations that typically arise in the first trimester of pregnancy, most commonly in the embryonic period: weeks 3–8 of post-conceptional age, which is equivalent to weeks 5–10 of gestational age (the number of weeks after the first day of the last menstrual period). For organ malformations the first trimester of pregnancy was thus considered the clinically relevant time period, while for other anomalies, such as congenital hypothyroidism, the relevant ‘window of exposure’ is less clear.

This review (PROSPERO no. CRD42020173519) forms part of a larger systematic mapping review on the impacts of high temperatures and adaptation interventions on human health (PROSPERO no. CRD42018118113) [5]. 

## 3. Results

The search in the mapping review identified 14,880 articles after removal of duplicate records (Figure 1). Overall, 2273 articles were classified as “heat-exposure studies or heat adaptation studies”, and 175 of these were selected for full text screening, with 13 eligible for this review [22,23,24,25,26,27,28,29,30,31,32,33,34]. A meta-analysis was not conducted due to considerable methodological diversity, statistical heterogeneity and the limited number of studies on each congenital defect. 

Six studies were set in the United States, four in Asia and the Middle East, two in Canada, one in Europe, and none in Central and South America or in Africa. The studies were published between 1991 and 2018 and covered births that took place in the period 1982 to 2015. In total, six studies reported on cardiac defects [22,25,28,30,32,33], three on neural tube defects [24,31,33], two on orofacial cleft or cranial defects [33,34], two on hypothyroidism [23,27], and one each on ocular anomalies [33], renal anomaly [33], hypospadias [29], musculoskeletal defects [33] and any lethal anomaly [26]. One paper covered multiple outcomes covering six organ systems [33]. Eleven studies reported measuring heat exposure in the first trimester, one of which extended to 14 weeks gestation [29]. Both studies on congenital hypothyroidism measured temperature exposure in the period shortly before childbirth [23,27].

### 3.1. Heat and Congenital Heart Anomalies

Of the six studies that assessed the effect of heat on cardiac defects, three reported adverse outcomes with heat exposure (Table 1). One study in Quebec, Canada [25] with more than 700,000 births examined the impacts of exposure for ≥10 days to ≥30 °C between 4 and 10 weeks gestation on different types of congenital cardiac anomalies. Associations were detected between the exposure and critical cardiac defects including transposition of great vessels, truncus arteriosus, and coarctation of the aorta, as well as noncritical cardiac defects including atrial septum defects (ASDs) and heterotaxy. Furthermore, the study showed that exposure to 15 days of ≥30 °C had larger effect sizes than shorter exposure periods, reaching a prevalence ratio of 1.37 for ASDs (95%CI = 1.10, 1.70). When individual weeks of gestation were examined separately, associations with elevated temperatures began in the fifth week gestation for ASDs and were strongest for multiple and noncritical defects towards the tenth week gestation, consistent with the highest risk period for cardiac development.

In another retrospective cohort study conducted in Israel during the cold season [22], the odds of multiple congenital heart defects increased 1.13-fold (95% CI odds ratio [OR] = 1.06, 1.21) and isolated ASDs 1.10-fold (95% CI = 1.02, 1.19) for each additional day of a heat wave during weeks three to eight of gestation. In an analysis using temperature exposure throughout the year, the odds of multiple defects were 1.03 higher per 1 °C increase in maximum daily peak temperature during weeks 3 to 8 (95% CI = 1.01, 1.05). 

A third study, a case-control study in eight states of the USA [30], reported higher rates of ventricular septal defects (VSDs) after heat waves (daily maximum temperature >90th percentile) lasting three to five days (ORs ranged from 2.17 to 2.57, all *p* < 0.05) at weeks 5 to 10 of gestation. Most other associations tested were not significant, but all point estimates were > 1.0. Additional associations were detected in analyses separated by states, including in New York state. There were no associations detected between high temperature and cardiac anomalies in three other reports [28,32,33]. The first examined the effects of heat exposure over the whole first trimester in New York State and found no associations [33] and the second, in the same location, reported an OR of a malformation of 1.27 (95% CI = 0.52, 3.13) in women who self-reported having had lengthy exposures to temperatures above 100°F (~37.8 °C), but no other findings suggesting an association [28]. The third study, set in Finland, found no impact of self-reported exposure during the first trimester to temperatures above 20 °C in the workplace [32]. 

### 3.2. Heat and Neural Tube Defects

A case-control study in New York state did not detect any temperature-outcome links, but point estimates for associations between heat waves and spina bifida without anencephalus (OR = 1.30; 95%CI = 0.82, 2.05) and microcephalus (OR = 1.10; 95% CI = 0.77, 1.58) were in the direction of effect [33] (Table 2). The exposure window used in the study covered a considerably larger time period than the critical period for neural tube defects (4–6 weeks gestation). A retrospective cohort study with about 900,000 live births over 24 years in Quebec, Canada identified few associations, aside from an increased odds of a neural tube defect with exposure to maximum temperatures above 30 °C during day 5 (OR = 1.56; 95% CI = 1.04, 2.35) and day 6 (OR = 1.49; 95% CI = 1.00, 2.21) of week 4 gestation [24]. The study excluded pregnancy terminations and stillbirths, and spanned the periods before and after national folic acid food fortification programs. By contrast, a population-based case-control study in 10 states of the USA did not observe significant associations between heat waves and neural tube defects in the overall population, but lower odds of a neural tube defect in some sub-analyses. For example, when analysis was restricted to the state of California, the odds ratio was 0.51 (95%CI = 0.28–0.93) with exposure to three consecutive days at temperatures >90th centile (98 °F) [31].

### 3.3. Heat and Orofacial Clefts and Craniofacial Defects

Two studies [30,33] examined the effect of heat exposure on orofacial clefts or craniofacial defects (Table 3). The first, a study in New York state, USA, found no association between maternal exposure to extreme heat events and cleft anomalies in overall analyses, with most point estimates around 1.00. In a sub-analysis stratified by ethnic group, however, cleft lip with or without cleft palate was 3.02 fold more common among the Hispanic population after heatwave exposure (95% CI = 1.44, 6.33), while no associations were noted in non-Hispanic groups [33]. A study in eight US states involving 907 cases reported null findings, with point estimates ranging from 0.45 to 1.43, and no discernable patterns in the findings [34].

### 3.4. Heat and Other Congenital Defects

Ocular disorders: A case-control study in New York state, reported a 1.51 fold higher odds of congenital cataract per 5 °F increase in mean daily minimum temperature (95% CI = 1.14, 1.99), and a nearly two-fold increase with heat waves, particularly between 4 and 7 weeks [33] (Table 3). In the same study, anopthalmia and micropthalmia were, however, less common at high mean minimum daily temperatures (OR = 0.71; 95% CI = 0.54, 0.94), with similar finds for daily mean and maximum temperature. 

Renal aplasia and hypoplasia: The same study in New York state as that described above noted a 1.17-fold increase in cases of renal agenesis or hypoplasia per 5 °F (~2.8 °C) increase in mean daily minimum temperature (95% CI = 1.00, 1.37) [33]. 

Hypospadias: One study in Turkey investigated whether higher ambient air temperature between 8 and 14 weeks gestation increases the risk of the anomaly [29]. Most cases occurred in the warm months and were 1.32-fold more likely during months with the highest mean monthly ambient temperatures than cooler months (95% CI = 1.08, 1.52). 

Gastroschisis: The same study in New York as discussed above found a reduction in gastroschisis following periods of heat waves (OR = 0.48; 95% CI = 0.28, 0.81), but no other significant heat-related impacts on other musculoskeletal anomalies [33]. 

Lethal malformations: A case-control study in Mexico reported a two-fold increase in anencephalic and other lethal malformations with mean temperatures above 18 °C within the fifth week of pregnancy (95% CI = 0.96, 4.15) compared with lower mean temperatures [26].

### 3.5. Heat and Congenital Hypothyroidism 

Two studies reported on the effects of heat exposure in the last few weeks of pregnancy on congenital hypothyroidism diagnosed shortly after childbirth (Table 4). A study in Iran found that the incidence of congenital hypothyroidism was lowest in the warmest months (July/August) where the average monthly temperature was reported to be 38.9 °C [23]. Cases of congenital hypothyroidism were 4% less likely for each 1 °C increase in temperature at the time of birth (95%CI = 2%, 6%) [23]. Likewise, a study in Japan demonstrated a negative linear association between the incidence of congenital hypothyroidism and a monthly average ambient temperature in the month of birth (r = −0.89). The highest incidence of congenital hypothyroidism occurred at the average temperature of 5.4 °C in January [27]. In that study, cases of congenital hypothyroidism due to thyroid dysgenesis which occurs during embryogenesis were not associated with temperature at childbirth, while cases due to other causes varied with temperature, with males most affected.

## 4. Discussion

While available data on impacts of heat exposure on pregnancy outcomes such as preterm birth have recently been reviewed, including by authors of this paper [5], this is the first paper to systematically assess exposure to high ambient temperatures and congenital anomalies. In total, only 13 studies were identified, which is disappointing given the public health implications of potential intersections between congenital anomalies and heat exposure. Though there was marked heterogeneity between studies in measurements of heat exposure, and the windows of gestation and types of anomalies examined, overall, across all the studies, a total of nine types of anomalies were identified as occurring more frequently during periods of high temperatures (ASD, VSD, multiple cardiac heart anomalies, spina bifida, orofacial clefts or cleft lip with cleft palate, renal agenesis or hypoplasia, hypospadias, congenital cataracts, and lethal malformations). Importantly, however, in some analyses the occurrence of some anomalies declined at higher temperatures, namely neural tube defects, congenital hypothyroidism, gastroschisis, and anopthalmia or micropthalmia [23,27,31,33]. Three studies reported non-significant findings [28,32,33], one of which examined multiple anomalies and had mixed results [33] while the other two used self-reported heat exposures [28,32]. 

Studies on heat exposure and cardiac defects are especially important as these defects are a large, emerging global problem and responsible for an estimated 260,000 deaths each year [35]. Only three of the six studies detected adverse congenital outcomes with heat exposures, while a fourth reported outcomes in the direction of effect. Differences between studies in climate zones may, in part, account for variations between the studies. For instance, the study from Israel [22] reported a correlation between heat and anomalies in the cold season, including a weak association with isolated ASDs but not VSDs. By contrast, a study in Quebec, Canada, [25] reported a significant association between exposure to temperatures ≥ 30 °C and ASDs. It is possible that in Israel people may acclimate to the warm Mediterranean type summers, but then be more vulnerable to unexpected hot periods during the cold season. By contrast, Quebec has long cold periods and its residents may not be acclimated to heat at any point in the year, and thus vulnerable to hot spells in any season. Of note, several of the studies that reported null findings were set in countries with temperate, sub-tropical or even polar climate zones where it may be rare for the temperatures to reach the threshold level required to cause a defect (assuming any links operate through a threshold mechanism) [28,32,33]. Overall, though it is clearly difficult to compare findings between studies, it appears that heat associations with ASDs were the largest and most consistent of all the anomalies assessed. 

Although human and animal studies suggest that the central nervous system is especially sensitive to temperature impacts through hyperthermia in a febrile episode, for example, [6,7,8,9,12,13], only one of the three studies included in our review reported an association between heat and neural tube defects [24]. Potentially, folate supplements in pregnancy or in fortified foodstuffs such as bread have attenuated these impacts [36]. This assertion is supported by evidence that folate supplements appear to reduce the teratogenic impacts of maternal fever [37,38] and the teratogenicity of heat exposure in mice [39]. Increases in global temperatures and carbon dioxide levels diminish the nutritional content of crops, providing further rationale for folate supplementation of food products and for women during pregnancy. Perhaps most importantly, associations between heat exposure and neural tube defects are difficult to interpret as many defects lead to early pregnancy spontaneous abortions, which may not be detected or recorded in registries. Large numbers of undetected early pregnancy abortions caused by environmental exposures can translate into a reduced number of fetal malformations at birth in exposed women and spurious conclusions about the exposure being protective. 

Consistent with the studies identified in this review, researchers in Japan [40], England [41], and Finland [42] reported that the incidence of congenital hypothyroidism was highest during the cold seasons. One study in Iran, however, showed a higher prevalence of congenital hypothyroidism during the warmer season [43]. These findings are difficult to interpret, however, since congenital hypothyroidism results from a range of aetiologies which operate at different periods of gestation and have diverse pathogenetic processes, including thyroid dysgenesis (a malformation of embryological development in early pregnancy), autosomal recessive genetic disorders, iodine deficiency, maternal autoantibodies, and anti-thyroid medications [44,45]. Causes vary geographically, especially those related to iodine deficiency and salt fortification, and autosomal recessive cases are more frequent in societies where consanguinity is common. In the study by Gu et al. which measured the impact of temperature exposure near to childbirth, the absence of an association between temperature and thyroid dysgenesis suggests that links between hypothyroidism and temperature may be mediated by factors other than heat impacts on embryo development [27]. None of the studies presented on congenital hypothyroidism specifically examined heat exposure during the embryological development period, which is a key gap in evidence. 

One study on the effect of heat on orofacial clefts or craniofacial defects only detected associations in a subgroup of Hispanic women [33], while the other had mixed findings, including protective effects in some analyses [34]. A case-control study [46] examining use of hot tubs in the first trimester and orofacial clefts did not find any associations, while, by contrast, increased risks were noted in women who reported using electrical bed-heating devices [47] or taking hot showers or baths [48]. 

## 5. Review Limitations

This review has several limitations. Considerable measurement error is likely and may under- or over-estimate associations. While studies involving animals can control the timing, duration and intensity of heat exposure, there are major challenges in accurately determining heat exposure in humans. Temperature measurements were operationalized in a multitude of ways and multiple testing may incur Type II errors. Many studies used the temperature at the site of birth as a proxy for the heat exposure of the woman during the first trimester of her pregnancy. This may be a poor proxy for actual exposures as many women may have lived some distance from the facility. The studies used outdoor or self-reported temperature measures as a proxy for actual heat exposure, which depends largely on the heat resistance of the built environment and the presence of air conditioning, for example. Additionally, sources of heat exposure such as saunas and hot tubs themselves implicated in birth defects in some studies [46,47,48], are difficult to measure and were not considered in the included studies. Further, there are challenges in accurately determining gestation, and thus the periods of embryogenesis relevant to each defect [49]. Several factors, such as air pollution and toxin exposure, teratogenic drug use, seasonal variations in infections, and in hormones in pregnancy, could confound or mediate heat impacts. Lastly, and perhaps most importantly, data on elective terminations, spontaneous abortions, or stillbirths related to congenital anomalies were not presented in most studies, making it difficult to draw conclusions about anomalies that have high rates of embryo or fetal demise, such as neural tube defects. Analyses of large registries of assisted conceptions may provide the only means of overcoming this limitation. 

## 6. Conclusions

From the available evidence, it appears that heat exposure may increase the risk of some birth defects, with evidence strongest for cardiac lesions such as ASDs and VSDs. The limited number of studies and heterogeneity of findingsmakes it difficult to draw definitive conclusions about the teratogenicity of heat exposure or size of such effects. Few studies were reported from resource-constrained settings, which have the highest burden of congenital anomalies and the least resources to treat long-term morbidities. Analyses of temperature exposure and outcomes using data in large registries of congenital anomalies, including those in low- and middle-income countries [50], could be done at relatively low cost. Studies providing more definitive evidence are especially pressing given the rapidly warming environment.

As temperatures escalate across the world, exposure to extreme heat in heatwaves is becoming more common, as are briefer periods of high temperatures, which generally account for the largest heat impacts. Even modest sized impacts of heat exposure would constitute an important attributable risk for congenital anomalies, a major concern given the mortality risks areoften lifelong sequelae of these conditions. 

Overall, this review, together with the large number of studies demonstrating the impacts of high temperatures on mortality and a range of morbidities, highlights the potential health threats posed by climate change for high-risk groups such as pregnant women. These concerns are especially pronounced among women in resource-constrained settings who have little or no access to interventions such as air conditioning. Global ‘thermal inequities’ are stark and widening. Increasing coverage of air conditioning in high-income countries, those largely responsible for climate change, provide high levels of protection against heat while people elsewhere face the full, mounting brunt of extreme heat. Globally, most attention to date on heat-health impacts has been focused on the elderly or those with chronic health conditions in wealthy countries. Attention of researchers and funders needs to shift to the individual and societal burdens of heat on other key populations such as pregnant women and, indeed, on embryos and fetuses. 

## Figures and Tables

**Figure 1 ijerph-18-04910-f001:**
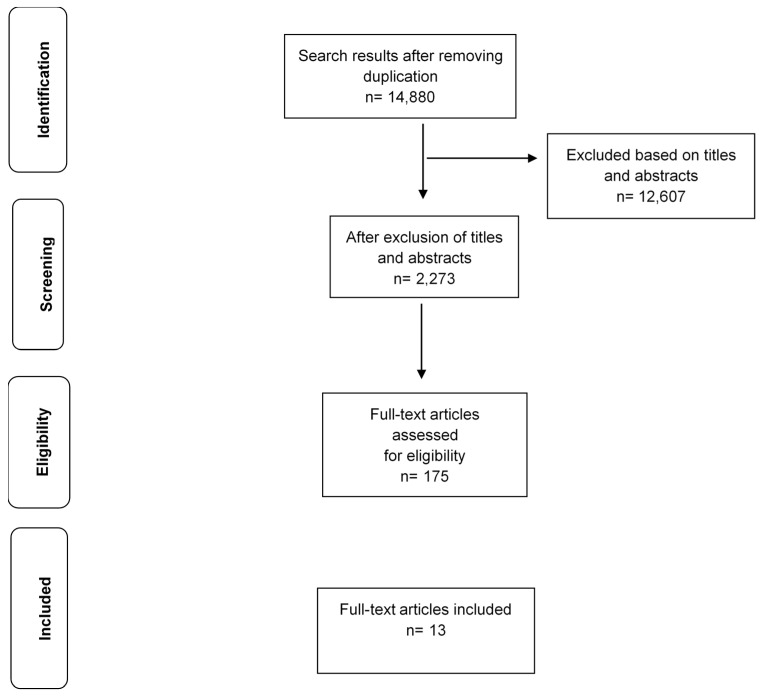
PRISMA flow diagram.

**Table 1 ijerph-18-04910-t001:** Associations between heat exposure and congenital heart anomalies.

Author (Year)	Country of Study	Number of Cases	Study Period	Time of Exposure Measurement	Controls or Comparator Group	Study Outcomes
Tikkanen and Heinonen, (1991) [32]	Finland	*n* = 573	1982–1984	First trimester	*n* = 1200	No association between self-reported exposure to temperatures during the first trimester of pregnancy ≥20 °C in the work environment and risk of cardiac malformation (*p* > 0.05)
Judge et al. (2004) [28]	New York state, USA	*n* = 502	1988–1991	1 month before pregnancy to date pregnancy diagnosed	*n* = 1066	Self-reported exposure to >100 °F (~38 °C) in early pregnancy (2.7% of women). OR of any cardiovascular anomaly=1.13 (95% CI = 0.59, 2.19) and >10 hours/week versus never OR = 1.27 (95% CI = 0.52–3.13)
Van Zutphen et al. (2012) [33] *	New York State, excluding New York City, USA	13 types of anomalies, n ranged from 9 with common truncus to 1579 with VSD	1992–2006 (summer months June–August)	First trimester	*n* = 59,328	No associations detected between mean and maximum universal apparent temperature, heat waves and days >90th centile, and cardiovascular defects.
Agay-Shay et al. (2013) [22]	Israel, Tel Aviv	*n* = 1630 (607 cases with multiple CHDs, 542 with isolated ASDs and 481 with isolated VSDs)	2000–2006	Weeks 3–8 (unclear if this refers to weeks post-conception or gestation)	*n* = 130,402	Whole year period. OR = 1.03 (95% CI = 1.01; 1.05) for multiple CHDs for exposure to maximum daily peak temperature (per 1 °C increase). Isolated ASD OR = 1.02 (95% CI = 1.00, 1.04) per 1 °C increase in average daily temperature. Quartile 3 temperature versus Q1 OR = 1.34 (95% CI = 1.06, 1.70), Q4 OR = 1.27 (95% CI = 1.00, 1.61). In the cold season exposure to the average ambient temperature and the maximum peak temperature (per 1 °C increase) increased the risk for multiple CHDs (OR = 1.05; 95% CI = 1.00, 1.10, and OR = 1.03, 95% CI = 1.01, 1.05, respectively). Comparing the highest to lowest quartiles of mean temperature increased the risk for multiple CHDs (OR = 1.41, 95% CI = 1.03, 1.94). 1-day increase in the extreme heat events showed increased risk for multiple CHDs (OR = 1.13, 95% CI = 1.06, 1.21) and also for isolated ASDs (OR = 1.10 95% CI = 1.02, 1.19). A 1-day increase in the extreme heat events based on the previous 90 days increased risk for multiple CHDs (OR = 1.02, 95% CI = 1.00, 1.04). VSD point estimates around 1.0, except per 1 °C increase in average daily temperature OR = 1.08 (95% CI = 1.00, 1.16)
Auger et al. (2017) [25]	Quebec, Canada	*n* = 6482 (*n* = 539 with critical heart defects and *n* = 5943 noncritical heart defects)	1988–2012 (summer months April –September)	Weeks 2–8 post-conception	*n* = 704,209	10 days ≥30 °C higher prevalence versus 0 days, of transposition of great vessels (29.2 vs. 19.2 per 100,000), truncus arteriosus (12.2 vs. 5.5 per 100,000), coarctation of aorta (21.9 vs. 16.5 per 100,000), ASD (413.2 vs. 289.0 per 100,000), defects of the aorta (19.4 vs. 11.9 per 100,000), heterotaxy (14.6 vs. 8.2 per 100,000), and other defects (255.2 vs. 223.0 per 100,000). Single and multiple defects also higher. Higher differences with longer exposure, especially with ASD, 15 days ≥ 30 °C (PR = 1.37, 95% CI = 1.10, 1.70). ASD associations highest in weeks 2 and 8. PR highest week 7, e.g., 32 °C associated with 1.13 times (95% CI: 1.01, 1.26) risk relative to 20°C. Maximum temperatures of 32 °C associated with multiple defects week 8 (PR = 1.31, 95% CI = 1.04, 1.65) compared with 20 °C.
Lin et al. (2018) [30]	USA 8 states	*n* = 5848 congenital heart defects, 4 types	1997–2007	Weeks 3–8 post-conception	*n* = 5742	Study examines ≥2 days with daily Tmax >95th centile (EHE95). ≥3 days with Tmax above the 90th percentile (EHE90). Duration of EHE90 or EHE95, n total days, and n consecutive days. Most associations null with overall defects, though all point estimates >1.0. VSD and ASD defects not significant, but almost all estimates >1.0, higher in Summer. VSD summer EHE95 OR = 1.18 (95% CI = 0.81–1.72). VSD spring EHE95 OR = 1.06 (95% CI = 0.41–2.74). ASD summer EHE95 OR = 1.32 (95% CI = 0.88–1.99). ASD spring EHE95 OR = 1.15 (95% CI = 0.33–4.04). VSD EHE90 durations of 3–5 days ORs ranged 2.17–2.57 all *p* < 0.05 in summer. OR point estimates generally increased with additional duration of exposure. Higher effect sizes in some regions, e.g., OR = 2.28 for EHE95 in Spring in New York for VSD and 1.87 (95% CI = 1.11, 3.16) for ASD and EHE95 duration. EHE95 total days and left ventricular outflow tract obstruction in Utah OR = 1.53 (95% CI = 1.00, 2.35), and septal defects in Iowa OR = 1.71 (95% CI = 1.09, 2.69). EHE95 duration and conotruncal defects in Utah OR = 1.34 (95% CI = 1.00, 1.81), septal defects in New York OR = 1.30 (95% CI = 1.05, 1.62). Association between temperature and VSD increased with magnitude and duration of high temperature exposure.

CI: confidence interval; OR: odds ratio; Studies listed in chronological order; ASD: atrial septum defect; CHD: congenital heart defect; EHEs: extreme heat events EHE90: defined as at least three consecutive days with daily maximum temperature above 90th percentile; UAT: universal apparent temperature; VSD: ventricular septal defects. * Study assessed defects in multiple organ systems, each presented in their respective tables.

**Table 2 ijerph-18-04910-t002:** Associations between heat exposure and neural tube defects.

Author (Year).	Country of Study	Number of Participants	Study Period	Time of Exposure Measurement	Controls or Comparator Group	Study Outcomes
Van Zutphen et al. (2012) [33] *	New York State, excluding New York City, USA	5 anomalies: anencephalus (*n* = 21), spina bifida without anencephalus (*n* = 114), hydrocephalus without spina bifida (311), encephalocele (*n* = 25) and microcephalus (*n* = 199)	1992–2006 (summer months June-August)	First trimester	*n* = 59,328	No association detected between mean and maximum universal apparent temperature, heat waves and days >90th centile and nervous system defects. Spina bifida without anencephalus OR = 1.12 with daily minimum apparent temperature (95% CI = 0.92, 1.35), OR = 1.30 with heat wave exposure, (95% CI = 0.82, 2.05). Microcephalus and heat waves exposure OR = 1.10 (95% CI = 0.77, 1.58).
Auger et al. (2017) [24]	Quebec, Canada	*n* = 297 neural tube defects, including spina bifida, anencephalus and encephalocoele	1988–2012 (April to September)	Weeks 3–4 post-conception	*n* = 887,710	Overall neural tube defects no pattern with maximum weekly temperature during the third- or fourth-week post-conception. Prevalence of spina bifida was higher for maximum weekly temperatures of ≥30°C during week 4 (29.5 per 100,000; 95% CI = 21.3, 37.8) versus 25.0 per 100 000 at 20–24.9 °C (95% CI 18.2 to 31.7); CIs wide. No pattern with anencephalus or encephalocoele. Compared to 20 °C, max daily temperature of 30 °C, OR = 1.56 of any neural tube defect on day 5 (95% CI = 1.04, 2.35) and OR = 1.49 on day 6 (95% CI = 1.00, 2.21) of week 4. Spina bifida and anencephalus or encephalocoele weak associations at end of critical exposure window. Associations with neural tube defects with higher temperatures towards the end of week 4.
Soim et al. (2017) [31]	USA 10 states	*n* = 326	1997–2007	Weeks 3–4 post-conception	*n* = 1781	Heat event (2 consecutive days daily apparent temperature max 95th (HE 95th) or 3 consecutive days at 90th centile (HE 90th). No associations detected between NTDs and HE 95th or 90th overall population. HE90 ((98°F) versus no HE90 in California OR = 0.51 (95% CI = 0.28–0.93); negative association. Overall population HE90 of 3 days duration versus 0 days OR = 0.66 (95% CI = 0.45–0.98) and OR = 0.33 (95% CI = 0.12–0.94) in California.

CI: Confidence interval; OR: odds ratio; HE heat event; Studies listed in chronological order; NTD: neural tube defect. * Study assessed defects in multiple organ systems, each presented in their respective tables.

**Table 3 ijerph-18-04910-t003:** Results of heat exposure and orofacial cleft or craniofacial, musculoskeletal, genitourinary, eye and lethal congenital anomalies.

Author (Year)	Country of Study	Number of Participants	Study Period	Time of Exposure Measurement	Controls or Comparator Group	Study Outcomes
Van Zutphen et al. (2012) [33] *	New York State, excluding New York City, USA	Three anomalies: choanal atresia (*n* = 99), cleft palate without cleft lip (*n* = 340), cleft lip ± cleft palate (*n* = 501)	1992–2006 (summer months June–August)	First trimester	*n* = 59,328	No association detected between mean and maximum universal apparent temperature, heat waves and days >90th centile, and craniofacial defects. Most point estimates around 1.00. With heatwave exposure, cleft lip with or without cleft palate in Hispanic (OR = 3.02; 95% CI = 1.44, 6.33) versus non-Hispanic (OR = 0.83; 95% CI = 0.67, 1.05).
Soim et al. (2018) [34]	USA 8 states	*n* = 907 live-born infants, stillbirths, and induced terminations with orofacial clefts or cleft lip with cleft palate	1997–2007	First 8 weeks post-conception	*n* = 2206	HE95: ≥2 consecutive days with apparent Tmax >95th centile HE90: ≥3 consecutive days apparent Tmax >90th centile. Point estimates range from estimates ranged from 0.45 to 1.43, no significant differences detected. In North Carolina, 3 days of HE95 OR = 1.89 (95% CI = 1.11, 3.23) versus 0 days. 4 days HE90 OR = 1.70 (95% CI = 1.02, 2.81). Iowa 3 days HE90 OR = 0.42 (95% CI = 0.22, 0.82).
Van Zutphen et al. (2012) [33] *	New York State, excluding New York City, USA	Congenital cataracts (*n* = 75), anophthalmia or microphthalmia (*n* = 34)	1992–2006 (summer months June–August)	First trimester	*n* = 59,328	A 5-degree Fahrenheit (2.8 °C) increase in the mean daily minimum universal apparent temperature (UAT) was associated with an increase in congenital cataracts (OR = 1.51; 95% CI = 1.14, 1.99). Congenital cataracts were also associated with heat waves (OR = 1.97; 95% CI = 1.17, 3.32), number of heat waves (OR = 1.45; 95% CI = 1.04, 2.02), and number of days above 90th centile (OR = 1.09; 95% CI = 1.02, 1.17). Associations between heat and congenital cataracts were positive for weeks 4–7, but not from week 10 onwards. Higher mean minimum UAT was associated with reduced anophthalmia or microphthalmia (OR = 0.71; 95% CI = 0.54, 0.94), as was daily mean UAT (OR = 0.70, 95% CI = 0.53, 0.93) and daily maximum (OR = 0.70, 95% CI = 0.52, 0.93)
Van Zutphen et al. (2012) [33] *	New York State, excluding New York City, USA	*n* = 174 with renal agenesis or hypoplasia	1992–2006 (summer months June–August)	First trimester	*n* = 59,328	A 5 °C Fahrenheit (2.8 °C) increase in mean daily minimum universal apparent temperature (UAT) associated with renal agenesis/hypoplasia (OR = 1.17; 95% CI = 1.00, 1.37). Though not significant, OR point estimate ranged from 1.13–1.15 with daily mean and maximum UAT and the outcome
Kilinc et al. (2016) [29]	Ankara and Istanbul, Turkey	*n* = 1709 with hypospadias	2000–2015	Weeks 8–14	*n* = 4946 other urological treatments	More cases occurred in warmer than colder months. Tmax monthly in cases = 36.4 ± 10.8 versus controls 26.0 ± 9.6 (*p* = 0.01) and mean monthly temperature, 22.5 ± 13.9 in cases versus 18.7 ± 11.3 in controls (*p* = 0.01). Mean and maximum monthly ambient temperatures in summer increased OR (OR = 1.32; 95% CI = 1.08, 1.52; and OR = 1.22; 95% CI = 0.99, 1.54), respectively
Van Zutphen et al. (2012) [33] *	New York State, excluding New York City, USA	Four musculoskeletal defects: upper limb reduction (*n* = 105), lower limb reduction (*n* = 85), gastroschisis (*n* = 108) and omphalocele (*n* = 81)	1992–2006 (summer months June–August)	First trimester	*n* = 59,328	No associations detected between mean and maximum universal apparent temperature, and musculoskeletal defects. However, OR of gastroschisis significantly decreased with heat wave events (OR = 0.48; 95% CI = 0.28, 0.81) and number of heat waves (OR = 0.63; 95% CI = 0.43, 0.92).
Davies (2000) [26]	Mexico City, Mexico	*n* = 38 lethal malformation deaths	1982–1984	Week 5	*n* = 335 other perinatal deaths no lethal malformations	OR = 2.0 of a perinatal death with a lethal malformation at a mean temperature >18 °C in week 5 versus perinatal deaths with a lethal malformation at a mean temperature <18 °C (95% CI = 0.96, 4,15; *p* = 0.04)

CI: confidence interval; OR: odds ratio; UAT: universal apparent temperature; Studies listed in chronological order. * Study assessed defects in multiple organ systems, each presented in their respective tables.

**Table 4 ijerph-18-04910-t004:** Associations between heat exposure and congenital hypothyroidism.

Author (Year)	Country of Study	Number of Participants	Study Period	Time of Exposure Measurement	Controls or Comparator Group	Study Outcomes
Gu et al. (2007) [27]	Japan 11 sites	*n* = 1586	1994–2003	Month of birth	0	Highest incidence at mean temperature of 5.4 °C. Temperature negatively correlated with incidence (−0.89, *p* < 0.001). Correlation highest in males.
Aminzadeh et al. (2010) [23]	Ahvaz, southwest Iran	*n* = 1131 had an abnormal TSH level*n* = 142 cases	2006–2008	Childbirth	*n* = 45,802	4% reduction per 1 °C increase in mean temperature at childbirth (OR = 0.96, 95% CI = 0.94, 0.98; *p* < 0.001). Similar with males and females. Lowest incidence in hottest month, highest in coldest. Linear relationship between the anomaly and temperature (r = 0.87, *p* < 0.001)

CI: confidence interval; OR: odds ratio; Studies listed in chronological order.

## Data Availability

Not Applicable.

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
