# Peer review of "Impacts of High Environmental Temperatures on Congenital Anomalies: A Systematic Review"

_ijerph, 2021, doi:10.3390/ijerph18094910_

Round 1
Reviewer 1 Report
To the Authors
The authors present a large literature review of maternal temperature exposure and the risk of congenital anomalies. They point out the disparity in studies and the need for more investigations in resource-poor areas of the world. I have no major comments.
Minor comments:
The first paragraph has an internal contradiction that is probably just semantics. The first sentence states a WHO estimate of newborn deaths due to congenital anomalies. The third sentence says that WHO doesn’t count child deaths due to congenital anomalies. The supposition is that it is that WHO does not directly count child deaths due to congenital anomalies, but based on XXX prevalence of anomalies,, and YYY annual infant deaths, it estimates that ### annual infant deaths are related to congenital anomalies.
Introduction. More cases in the summer may also relate to more babies being born in the summer. At least in the Northern Hemisphere the high birth months are August and September, which is late summer. Is “cases” an absolute number or a prevalence rate?
Another confounder is maternal mobility: her presence in the delivery location does not necessarily translate to her location there at the relevant time of gestation for the defect. Studies that looked at ambient temperature instead of maternal report have that problem. Of course, maternal recall is another problem all together.
Author Response
Reviewer 1
To the Authors
Comment: The authors present a large literature review of maternal temperature exposure and the risk of congenital anomalies. They point out the disparity in studies and the need for more investigations in resource-poor areas of the world. I have no major comments.
Response: Many thanks for the review and fast response.
Minor comments:
Comment: The first paragraph has an internal contradiction that is probably just semantics. The first sentence states a WHO estimate of newborn deaths due to congenital anomalies. The third sentence says that WHO doesn’t count child deaths due to congenital anomalies. The supposition is that it is that WHO does not directly count child deaths due to congenital anomalies, but based on XXX prevalence of anomalies,, and YYY annual infant deaths, it estimates that ### annual infant deaths are related to congenital anomalies.
Response: This is indeed semantics, and confusing. The text is correct in that the WHO figures do not include deaths from congenital anomalies that occur in older children, and only include newborn deaths. I removed the phrase ‘child deaths’ as this is confusing, the key deaths that are not counted are the spontaneous abortions and stilbirths.
Comment Introduction. More cases in the summer may also relate to more babies being born in the summer. At least in the Northern Hemisphere the high birth months are August and September, which is late summer. Is “cases” an absolute number or a prevalence rate?
Response: This is a very good point, I have never considered this. Indeed ‘seasonal’ effect does warrant scrutiny. And seasons of birth versus season of conception need to be taken into account. I have removed this from the introduction. I left the report of seasons when discussing congenital hypothyroidism in the discussion as there the caveats of seasonality are set out. In those four studies the incidence of hypothyroidism is presented, not number of cases, so the limitation mentioned does not apply here.
Comment: Another confounder is maternal mobility: her presence in the delivery location does not necessarily translate to her location there at the relevant time of gestation for the defect. Studies that looked at ambient temperature instead of maternal report have that problem. Of course, maternal recall is another problem all together.
Response: As the reviewer points out, there are potential biases in exposure measurement. The problem of mobility is perhaps more pronounced when the ‘lag’ period is 6-8 months. I have added this to the limitations section. The concern about self-report is already mentioned, though these studies provide some useful information I think, if taken in conjunction with other evidence. We added the following text: ‘Many studies used the temperature at the site of birth as a proxy for the heat exposure of the woman during the first trimester of her pregnancy. This may be a poor proxy for actual exposures as many women may have lived some distance from the facility.’
Reviewer 2 Report
In this interesting study, the authors summarized the existing evidence on the association between exposure to environmental heat and increased risk of congenital anomalies. The manuscript is well written, and conclusions are consistent with the reported results. I point out a few comments that should be addressed before the manuscript can be considered for publication.
Methods
Line 114. “We only included studies on humans, published in English, German or Italian”. Why did the authors also choose German and Italian, two languages respectively spoken exclusively in Germany and Italy?
Why did the authors decide to include congenital hypothyroidism among the defects analyzed in the systematic review? Congenital hypothyroidism is a rare disease rather than a congenital anomaly, in fact it is not comprised in the list of anomalies in EUROCAT (https://eu-rd-platform.jrc.ec.europa.eu/eurocat/eurocat-data/prevalence_en), although it may be associated with congenital malformations (Rastogi 2010 “Congenital hypothyroidism”).
Results
Table 2, last column. “Spina bifida and anencephalus or encephalocoele weak associations at end of critical exposure window.” Please add “the” before “end”.
Table 2, last column. “Associations with neural tube defects with higher temperatures towards the end of week 4.” Please replace “with neural tube defects” with “of neural tube defects”.
Author Response
Reviewer 2
Comment: In this interesting study, the authors summarized the existing evidence on the association between exposure to environmental heat and increased risk of congenital anomalies. The manuscript is well written, and conclusions are consistent with the reported results. I point out a few comments that should be addressed before the manuscript can be considered for publication.
Response: Many thanks for the review and kind comments.
Methods
Comments: Line 114. “We only included studies on humans, published in English, German or Italian”. Why did the authors also choose German and Italian, two languages respectively spoken exclusively in Germany and Italy?
Response: This is a good point. The review team included German and Italian speaking people, so it was a question of convenience, rather than a scientific choice. While a broader range of languages would have been preferable, we feel it is useful to include as many as are possible within the resources available. We acknowledge the bias towards English and EU literature.
Comment: Why did the authors decide to include congenital hypothyroidism among the defects analyzed in the systematic review? Congenital hypothyroidism is a rare disease rather than a congenital anomaly, in fact it is not comprised in the list of anomalies in EUROCAT (https://eu-rd-platform.jrc.ec.europa.eu/eurocat/eurocat-data/prevalence_en), although it may be associated with congenital malformations (Rastogi 2010 “Congenital hypothyroidism”).
Response: The reviewer has an important point here and the aetiology is complex. We used the WHO definition of a congenital anomaly, which would encompass congenital thyroid. ‘Congenital anomalies are also known as birth defects, congenital disorders or congenital malformations. Congenital anomalies can be defined as structural or functional anomalies (for example, metabolic disorders) that occur during intrauterine life and can be identified prenatally, at birth, or sometimes may only be detected later in infancy, such as hearing defects. In simple terms, congenital refers to the existence at or before birth.’ https://www.who.int/news-room/fact-sheets/detail/congenital-anomalies
Results
Comment: Table 2, last column. “Spina bifida and anencephalus or encephalocoele weak associations at end of critical exposure window.” Please add “the” before “end”.
Response: Many thanks for noticing this error, we have made the correction
Comment: Table 2, last column. “Associations with neural tube defects with higher temperatures towards the end of week 4.” Please replace “with neural tube defects” with “of neural tube defects”.
Response Many thanks for noticing this error, we have made the correction